# SAGE: Scalable Ground Truth Evaluations for Large Sparse Autoencoders

## Abstract

A key challenge in interpretability is to decompose model activations into meaningful features. Sparse autoencoders (SAEs) have emerged as a promising tool for this task. However, a central problem in evaluating the quality of SAEs is the absence of ground truth features to serve as an evaluation gold standard. Current evaluation methods for SAEs are therefore confronted with a significant trade-off: SAEs can either leverage toy models or other proxies with predefined ground truth features; or they use extensive prior knowledge of realistic task circuits. The former limits the generalizability of the evaluation results, while the latter limits the range of models and tasks that can be used for evaluations. We introduce SAGE: **S**calable **A**utoencoder **G**round-truth **E**valuation, an evaluation framework for SAEs that enables obtaining high-quality feature dictionaries for diverse tasks and feature distributions without relying on prior knowledge. Specifically, we lift previous limitations by showing that ground truth evaluations on realistic tasks can be automated and scaled. First, we show that we can automatically identify the cross-sections in the model where task-specific features are active. Second, we demonstrate that we can then compute the ground truth features at these cross-sections. Third, we introduce a novel reconstruction method which significantly reduces the amount of trained SAEs needed for the evaluation. This addresses scalability limitations in prior work and significantly simplifies the practical evaluations. We validate our results by evaluating SAEs on novel tasks on Pythia70M, GPT-2 Small, and Gemma-2-2B, thus demonstrating the scalability of our method to state-of-the-art open-source frontier models. These advancements pave the way for generalizable, large-scale evaluations of SAEs in interpretability research.

## 1 Introduction

Large language models (LLMs) have demonstrated remarkable performance across a wide variety of tasks (Brown et al., 2020; Raffel et al., 2020; Devlin et al., 2019; Radford et al., 2019; Vaswani et al., 2017). As a result they are increasingly deployed in high stake domains, for example healthcare (Luo et al., 2022), law (Shaheen et al., 2020), and finance (Li et al., 2023). However, the deployment of LLMs in such critical areas raises significant safety and ethical concerns (Bengio et al., 2024; Anderljung et al., 2023; Hendrycks & Mazeika, 2022), as their decisions can have profound consequences. This makes it crucial to understand the internal mechanisms and reasoning processes of these models, to ensure that their outputs are reliable, transparent, and aligned with human values.

A key challenge in interpreting LLMs is to decompose their internal activations into meaningful and interpretable features. Recently, sparse autoencoders (SAEs) have emerged as a promising solution for this task (Black et al., 2022; Cunningham et al., 2023). The quality of SAEs relies heavily on a range of hyperparameters such as the dictionary size, the sparsity constraint, and the choice of activation function (Gao et al., 2024). The dictionary size for instance influences the granularity of features found. This makes it important to strike the right balance between being too broad, which risks overseeing relevant features; or too specific, generating redundant or overly fragmented features. Therefore, realistic SAE evaluations—those that assess models on their trained tasks and within their application context—are critical to ensure that SAEs generalize well beyond synthetic benchmarks and perform reliably in real-world scenarios.

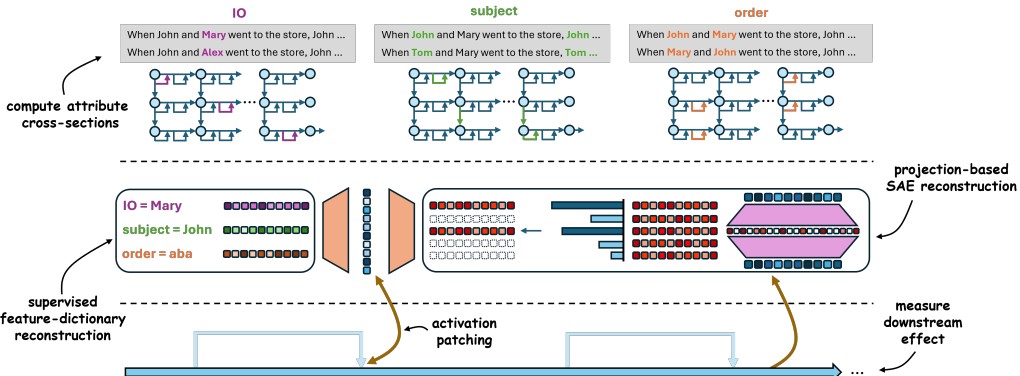

Figure 1: A graphical overview of SAGE: **S**calable **A**utoencoder **G**round-truth **E**valuation. SAGE scales SAE ground truth evaluation using three key steps: First, we identify attribute cross-sections, which are the relevant model components responsible for processing specific task-related attributes. Second, we compute supervised feature dictionaries that approximate ground truth features for our task. Third, we manipulate model activations using the supervised feature dictionary and an SAE, and evaluate the downstream effects to assess the SAE's quality against the approximate ground truth. We introduce a novel projection-based reconstruction using residual stream SAEs, which significantly reduces training overhead compared to prior art.

A central obstacle in evaluating SAEs in realistic settings is the lack of ground truth features that can serve as a gold standard for comparison. This leads to a problematic trade-off: researchers either rely on toy models with predefined ground truth features (Karvonen et al., 2024; Sharkey et al., 2023), which raises concerns about generalizability to real-world tasks; or they employ more realistic tasks that require extensive prior knowledge of the task circuit to derive ground truth features (Makelov et al., 2024), limiting scalability to other tasks or models. As a result, task-specific ground truth evaluations are currently limited to smaller models, such as GPT-2 Small, where well-studied tasks like indirect object identification (IOI) (Wang et al., 2023) are available. Nevertheless, approximate ground truth features have been identified in larger LLMs, including features for refusal (Arditi et al., 2024), sentiment (Tigges et al., 2023), and toxicity (Turner et al., 2024). Moreover, recent advances in automated circuit discovery methods have shown promise in automatically and efficiently identifying task-specific circuits in large LLMs (Conmy et al., 2023; Syed et al., 2023; Nanda, 2023). This suggests that ground truth evaluations of SAEs may scale to tasks without prior knowledge and to large frontier models.

Another challenge in scaling SAE evaluations to large frontier models is the need to train multiple SAEs across various model components (such as the query, key, and value vectors of attention heads) to fully capture the structure of the task circuit (Makelov et al., 2024). This becomes especially problematic for larger models, where the computational cost and extensive hyperparameter tuning required for training a family of SAEs make this approach infeasible at scale.

In this work, we introduce SAGE: **S**calable **A**utoencoder **G**round-truth **E**valuation (see Figure 1), an evaluation framework for SAEs that enables obtaining high-quality feature dictionaries for diverse tasks and feature distributions without relying on prior knowledge. Our method scales efficiently to large frontier models and their associated SAEs. Specifically, we:

1. demonstrate that automated circuit discovery methods can find model components where task-specific features are linearly represented. This allows for SAE evaluations on new tasks without requiring manual experimentation or prior knowledge.

2. show that supervised feature dictionaries derived from these components can serve as high-quality ground truths for evaluating SAEs, enabling a reliable comparison standard for the evaluation.

3. propose a new method for reconstructing sublayer activations, reducing the evaluation process to residual stream SAEs without compromising precision or generalizability. This

innovation makes it feasible to scale task-specific SAE evaluations to large models by drastically cutting the training overhead.

We apply this framework to existing evaluation benchmarks, validating the quality of our approach on models such as Pythia 70M (Biderman et al., 2023), GPT-2 Small (Radford et al., 2019), and GEMMA-2-2B (et al., 2024). Our results demonstrate that our method scales to large, state-of-the-art frontier LLMs and SAEs. With this approach, we enable researchers to generate task-specific evaluations for arbitrary SAEs, models, feature distributions, and tasks in a more efficient manner, paving the way for principled advancements in sparse dictionary learning and large-scale model interpretability.

## 2 RELATED WORK

In this section, we highlight some key contributions to the landscape of SAE evaluation methods.

**Toy Model Evaluations**   Sharkey et al. (2023) explore SAE evaluation using synthetic datasets with predefined ground truth features to assess how well SAEs recover features from superposition. Their findings demonstrate that a simple SAE with an L1 penalty can effectively separate features in a controlled, toy model setup, offering a useful benchmark for understanding feature recovery. Karvonen et al. (2024) propose evaluating SAEs using board games like chess and Othello, leveraging the clear structure of these games to pre-define interpretable features. They introduce two metrics: Board Reconstruction (the ability to reconstruct game states) and Coverage (the proportion of predefined features correctly captured). While these approaches provide straight-forward evaluations based on ground truth features, they are domain-specific and may not generalize well to other tasks and realistic models.

**Task-specific SAE Evaluation**   Makelov et al. (2024) present a principled framework for evaluating SAEs on GPT-2 Small using the IOI task. They compare several unsupervised SAEs with supervised feature dictionaries to assess their ability to approximate, control, and interpret model computations. Although this method offers a realistic, task-specific evaluation, it depends on prior knowledge of the IOI task and is therefore specific for the feature distribution of IOI and GPT-2 Small, limiting its scalability to other tasks and models.

**RAVEL Benchmark**   Huang et al. (2024) introduce RAVEL, a benchmark for evaluating activation disentanglement methods, including SAEs, in LLMs. RAVEL assesses the ability to isolate distinct features in polysemantic neurons through *interchange interventions*. RAVEL's Disentangle score measures whether a feature influences a target attribute without altering unrelated attributes, providing a comprehensive evaluation of feature disentanglement. A limitation of RAVEL is that its intervention sites are limited to the residual stream above the last entity token, potentially missing distributed representations across multiple tokens or layers.

## 3 BACKGROUND

**Tasks**   A task in the context of LLMs is a set of input prompts $\{p_k\}_{k \in N}$, which are fully defined over a set of attributes $\{a_i\}_{i \in I}$ that take values in $\{S_i\}_{i \in I}$, and a set of outputs $\{y_i\}_{k \in N}$. We say an LLM completes the task when the model predicts the outputs of the input prompts with a sufficient accuracy. Therefore, a "realistic task" for a SAE is a task that is completed by the LLM that the SAE has been trained on.

**Sparse Autoencoders**   A Sparse Autoencoder consists of two main components: an encoder and a decoder. The encoder maps the input data to a higher-dimensional latent space, while the decoder reconstructs the input from this latent space representation. Mathematically, let $\mathbf{x} \in \mathbb{R}^d$ represent the input vector (for example a residual stream component, an attention head output etc.), where $d$ is the input dimension and $m$ is the dimension of the latent space.

The encoder function is defined as:

$$\mathbf{c} = f(\mathbf{W}_e \mathbf{x} + \mathbf{b}_e) \tag{1}$$

where $\mathbf{W}_e \in \mathbb{R}^{m \times d}$ is the weight matrix of the encoder, $\mathbf{b}_e \in \mathbb{R}^m$ is the bias vector, $f$ is an element-wise nonlinearity and $\mathbf{c} \in \mathbb{R}^m$ represent the coefficients of feature activations living in the latent space with $m > d_{model}$.

The decoder reconstructs the input from the latent space representation, by multiplying the decoder weight matrix, consisting of the feature directions, with the computed feature coefficients:

$$\hat{\mathbf{x}} = \mathbf{W}_d \mathbf{c} + \mathbf{b}_d \tag{2}$$

where $\mathbf{W}_d \in \mathbb{R}^{d \times m}$ is the weight matrix of the decoder, $\mathbf{b}_d \in \mathbb{R}^d$ is the bias vector, and $\hat{\mathbf{x}}$ is the reconstructed input.

The learning objective of a Sparse Autoencoder is to minimize the $l_2$ reconstruction error while enforcing sparsity in the coefficients using an $l_1$ regularization term. The overall loss function for training a Sparse Autoencoder therefore combines the reconstruction loss and the sparsity loss:

$$L(\mathbf{x}, \hat{\mathbf{x}}, \mathbf{c}) = \|\mathbf{x} - \hat{\mathbf{x}}\|_2^2 + \alpha \|\mathbf{c}\|_1 \tag{3}$$

**Supervised feature dictionaries**  Supervised feature dictionaries, introduced by Makelov et al. (2024), are feature dictionaries that aim to capture the ground truth features of a specific task. In contrast to SAEs they are computed in a supervised manner and provide a structured way to disentangle and reconstruct the internal representations of LLMs. As they capture approximate ground truth features they can serve as a gold standard in evaluating SAEs. We describe the procedure to compute supervised feature dictionaries from Makelov et al. (2024) in Appendix A.1.

To obtain a supervised feature dictionary for a task, one needs to define the relevant attributes $\{a_i\}_{i \in I}$ of a task, as well as the values $\{S_i\}_{i \in I}$ that the attributes can take. The IOI task for example can be described with an *indirect object (io)* attribute, a *subject* attribute, as well as an *order* attribute, which described the ordering of the *io* and *subject* attributes. Given an internal activation $\mathbf{a}(p)$ over a task prompt $p$, and the associated supervised feature dictionary for the task, one can reconstruct the activation $\mathbf{a}(p)$ using

$$\mathbf{a}(p) \approx \mathbb{E}_{p \sim \mathcal{D}}[\mathbf{a}(p)] + \sum_{i \in I} \mathbf{u}_{a_i = v} := \hat{\mathbf{a}} \tag{4}$$

where $\hat{\mathbf{a}}$ is the reconstruction of $\mathbf{a}(p)$, and $\mathbf{u}_{a_i = v} \in \mathbb{R}^d$ is a feature corresponding to the $i$-th attribute having value $v \in S_i$.

In this formulation the supervised features are not weighted with coefficients. Makelov et al. (2024) argue that this works well for the IOI task, as name-features work like binary on-off switches. However, in the general case that does not hold, therefore we propose to compute weights to minimize the MSE loss between the reconstruction and the activation:

$$\lambda^* = \arg\min_{\lambda} \left\| \mathbf{a}(p) - \left( \mathbb{E}_{p \sim \mathcal{D}}[\mathbf{a}(p)] + \sum_{i \in I} \lambda_i \mathbf{u}_{a_i = v} \right) \right\|_2^2, \tag{5}$$

where $\lambda_i$ are the optimal weights for the reconstruction. This can be computed using the closed-form solution:

$$\lambda^* = \left( V^T V \right)^{-1} V^T \mathbf{a}(p), \tag{6}$$

where $V$ is the matrix whose columns are the supervised feature vectors.

**Circuit Discovery**  Circuit discovery methods aim to identify a subgraph of edges in a model's computational graph that solves a given task in an understandable manner (Wang et al., 2023). The process involves quantifying the causal importance of computational edges using intervention techniques.

Let $\mathbf{x}_{\text{clean}}$ be a clean input, $\mathbf{x}_{\text{corr}}$ a corrupt input, $L$ a metric (e.g., logit difference), and $E$ a computational edge between an upstream and downstream component in the transformer. The causal importance of edge $E$ is quantified by:

$$L(\mathbf{x}_{\text{clean}} | \text{do}(E = e_{\text{corr}})) - L(\mathbf{x}_{\text{clean}}) \tag{7}$$

where $\text{do}(E = e_{\text{corr}})$ denotes corrupting edge $E$. Corrupting an edge means that only the interaction between the upstream and downstream components is affected: the activation of the upstream

component under $\mathbf{x}_{\text{clean}}$ is replaced by its activation under $\mathbf{x}_{\text{corr}}$ at the point where the downstream component processes this activation, while other computations remain unchanged.

This manual approach is computationally intensive, requiring two forward passes for each edge (one for the corrupt activation, one for patching). To address this, Nanda (2023) introduced "attribution patching", which uses a first-order Taylor expansion to approximate the patching effect:

$$L(\mathbf{x}_{\text{clean}}|\text{do}(E = e_{\text{corr}})) - L(\mathbf{x}_{\text{clean}}) \approx (e_{\text{corr}} - e_{\text{clean}}) \cdot \frac{\partial L(\mathbf{x}_{\text{clean}}|\text{do}(E = e_{\text{clean}}))}{\partial e_{\text{clean}}} \tag{8}$$

This method requires only two forward passes (for clean and corrupt activations) and one backward pass (for gradients) to approximate the patching effect for *every computational edge* in the graph. Syed et al. (2023) demonstrated that this approach is not only more efficient than manual patching, but can outperform traditional patching techniques in circuit discovery. For edges in the model, the activations are obtained based on the upstream activation, and the gradients of the edge are obtained based on the downstream component.

However, attribution patching can face issues with zero gradients: Attribution patching uses a linear approximation of the change of metric $L$, however there is no guarantee that the metric $L$ changes linearly with respect to a specific activation in the model. Integrated gradients (Sundararajan et al., 2017), a method that tackles this problem by taking into account several gradients of intermediate activations between the clean and corrupt activation, improves the approximation results (Marks et al., 2024; Hanna et al., 2024):

$$\Delta L \approx (e_{\text{corr}} - e_{\text{clean}}) \cdot \frac{1}{m} \sum_{k=1}^{m} \frac{\partial L(\mathbf{x}_{\text{clean}}|\text{do}(E = e_{\text{clean}} + \frac{k}{m}(e_{\text{corr}} - e_{\text{clean}})))}{\partial e_{\text{clean}}} \tag{9}$$

where $\Delta L = L(\mathbf{x}_{\text{clean}}|\text{do}(E = e_{\text{corr}})) - L(\mathbf{x}_{\text{clean}})$.

For scenarios where the counterfactual differs only in small, causally specific aspects, Hanna et al. (2024) showed that the computationally cheaper clean-corrupt method performs comparably:

$$\Delta L \approx (e_{\text{corr}} - e_{\text{clean}}) \cdot \left( \frac{1}{2} \frac{\partial L(\mathbf{x}_{\text{clean}}|\text{do}(E = e_{\text{clean}}))}{\partial e_{\text{clean}}} + \frac{1}{2} \frac{\partial L(\mathbf{x}_{\text{corr}}|\text{do}(E = e_{\text{corr}}))}{\partial e_{\text{corr}}} \right) \tag{10}$$

This clean-corrupt method is therefore employed in this work.

## 4 METHODOLOGY

In this section, we describe our approach to scaling supervised feature dictionary evaluations.

**Discovering Cross-Sections**  The cross-sections of a circuit are the locations in the computational graph where a task-relevant feature is active. Therefore, supervised feature dictionaries are obtained at these locations to capture the approximate ground truth features. Cross-sections are also critical for conducting the actual evaluation, as manipulating the activations at these points is expected to have the most significant downstream effect on task performance, allowing us to assess the impact of task-relevant features on the overall functionality of the LLM. As Makelov et al. (2024) derives the cross-sections for the IOI task based on prior work of Wang et al. (2023), this does not work in the general case.

We propose to use attribution-based circuit discovery methods, to find the *attribute cross-sections*, the cross-sections for each attribute of the task, using the following procedure: Consider an attribute $\mathbf{a}_i$. To find the relevant cross-sections for the attribute, we sample pairs of inputs for the task $(x_{\text{clean}}, x_{\text{corr}})$ where each $x_{\text{corr}}$ differs in the value of the attribute $a_i$ compared to $x_{\text{corr}}$. Then we perform a forward pass and a backward pass with respect to the metric $L$ (for example logit difference) for each $x_{\text{clean}}$ and $x_{\text{corr}}$ and cache the activations and gradients of all components in the computational graph. Then we can obtain an approximation of the patching effect with equation 10 for each computational edge in the model and each input pair. We average the score of each edge over all input pairs, to get a robust approximation for the cross-sections across many examples of the task. The average score gives us an approximation for the effect on the logit difference, when patching in an activation of an example that differs only in attribute $a_i$. Thus, a large score for an edge indicates

that the feature $\mathbf{u}_{a_i=v}$ is active in the activation of the upstream component and processed by the downstream component.

We obtain a group of cross-sections for each attribute in $\{a_i\}_{i \in I}$ by taking the top $n$ edges with the largest average attribution score. Then, we separate the cross-sections with a positive attribution score, and those with a negative attribution score. This is because we want to evaluate SAEs and the supervised feature dictionary based on their downstream effect on the logits, thus we want cross-section groups with a similar effect.

**Derive Supervised Feature Dictionaries** After we found the attribute cross-sections for which we expect our desired features to be linearly represented in the activations of the upstream components, we obtain the supervised feature dictionaries.

**Projection-Based Reconstruction with Residual Stream SAEs** To evaluate SAEs in the context of a specific task, we need to apply them to the task's cross-sections to find the task-specific features. Previous work by Makelov et al. (2024) directly used the relevant key, query, and value vectors of attention heads in the IOI circuit as cross-sections. Therefore, for the evaluation they required a trained SAE for each key, query, and value subspace of each attention head that they use as a cross-section. Training these SAEs requires extensive training and challenging hyperparameter tuning. In a first step, to address this training overhead, we only apply SAEs and supervised feature dictionaries to the upstream components of our cross-sections, typically the attention head output. However, this still requires a trained SAE for each attention head that is part of a cross-section. Therefore, we propose a reconstruction method based solely on residual stream SAEs, allowing evaluations with only a handful of residual stream SAEs.

Consider a residual stream SAE with encoder weights $\mathbf{W}_e$ and decoder weights $\mathbf{W}_d$. The residual stream at layer $l$ can be expressed recursively as:

$$\mathbf{x}^{(l)} = \mathbf{x}^{(l-1)} + \text{Layer}_{l-1}(\mathbf{x}^{(l-1)})$$

This relationship allows us to express the residual stream at layer $l$ as a sum of all previous layer outputs:

$$\mathbf{x}^{(l)} = \mathbf{x}^{(0)} + \sum_{j=1}^{l} \text{Layer}_{j-1}(\mathbf{x}^{(j-1)}),$$

where $\mathbf{x}^{(0)}$ is the output of the embedding and positional encoding.

Each layer's output, $\text{Layer}_j(\mathbf{x}^{(j)})$, can be further decomposed into the outputs of attention heads and MLP components:

$$\text{Layer}_j(\mathbf{x}^{(j)}) = \sum_{k=1}^{H} \mathbf{h}_k(\mathbf{x}^{(j)}) + \text{MLP}(\mathbf{x}^{(j)}).$$

Thus, the residual stream at any layer $l$ can be represented as:

$$\mathbf{x}^{(l)} = \mathbf{x}^{(0)} + \sum_{j=1}^{l} \left( \sum_{k=1}^{H} \mathbf{h}_k(\mathbf{x}^{(j)}) + \text{MLP}(\mathbf{x}^{(j)}) \right).$$

The SAE is trained on this linear combination of attention head and MLP outputs. Therefore, if a specific attention head $h_k$ at layer $j$ writes in an important (i.e. task-specific) direction on the residual stream at layer $l$, the residual stream SAE must reconstruct this direction to reconstruct the residual stream at layer $l$.

Applying the residual stream encoder at layer $l$ yields:

$$\mathbf{c}^{(l)} = f(\mathbf{W}_e \mathbf{x}^{(l)} + \mathbf{b}_e)$$

where $\mathbf{c}^{(l)}$ are the coefficients of the SAE features for the residual stream at layer $l$.

The decoder reconstruction $\hat{\mathbf{x}}^{(l)}$ is given by:

$$\hat{\mathbf{x}}^{(l)} = \mathbf{W}_d \mathbf{c}^{(l)} + \mathbf{b}_d = \sum_{i=1}^{m} \mathbf{f}_i \mathbf{c}_i^{(l)} + \mathbf{b}_d$$

where $\mathbf{f}_i$ is a feature in $\mathbf{W}_d$ and $m$ is the dimension of the SAE.

To obtain the reconstruction for a sublayer activation $\mathbf{h}_k(\mathbf{x}^{(j)})$, we measure the alignment between each active feature $\mathbf{f}_i$ (i.e. $\mathbf{c}_i^{(l)} > 0$) and $\mathbf{h}_k(\mathbf{x}^{(j)})$ using the dot product:

$$\alpha_i = \mathbf{h}_k(\mathbf{x}^{(j)}) \cdot \mathbf{f}_i$$

We select all features for the reconstruction of $\mathbf{h}_k(\mathbf{x}^{(j)})$ for which $\alpha_i$ is greater than or equal to the average across all $\alpha$ (alternatively one could use a fixed threshold), yielding a set of features $\{f_t\}_{t \in T}$ that are represented in a similar direction as $\mathbf{h}_k(\mathbf{x}^{(j)})$. To obtain coefficients $\{c_t\}_{t \in T}$ for a good reconstruction of the sublayer activation, we minimize the mean squared error:

$$\left\| \mathbf{h}_k(\mathbf{x}^{(j)}) - \sum_{t=1}^{T} \mathbf{f}_t \mathbf{c}_t \right\|^2$$

This leads to the closed-form solution:

$$\mathbf{c} = (\mathbf{F}^\top \mathbf{F})^{-1} \mathbf{F}^\top \mathbf{h}_k(\mathbf{x}^{(j)})$$

where $\mathbf{F} \in \mathbb{R}^{d \times T}$ is the matrix of selected features and $\mathbf{c} \in \mathbb{R}^T$ is the vector of coefficients.

Based on our procedure of discovering cross-sections, we make the assumption that the sublayer activations of the upstream components of the cross-sections contain linear representations of task-relevant features $\mathbf{u}_{a_i=v}$ that have a significant effect on the logits. Thus, a good residual stream SAE for the task has to reconstruct this feature. Our method aims to find these features through the alignment filter process and provide a good reconstruction of the sublayer activation with respect to the task if the SAE is suitable for the task. With this approach we are able to perform edits on the task circuit with the same precision as previous methods, while significantly reducing the training overhead.

**Evaluations**  Based on the cross-sections, supervised feature dictionaries and projection-based SAE reconstruction, one can perform downstream tests on the SAEs and supervised feature dictionaries. In the scope of this work we aim to validate that our supervised feature dictionaries capture high quality approximations of ground truth features. Therefore, we reimplement Test 1 and Test 2 from Makelov et al. (2024) to evaluate our supervised feature dictionaries against several open-source SAEs across various models. As supervised feature dictionaries pass test 3 (interpretability test) tautologically, we do not reimplement the test in the scope of this work, but an evaluation based on this test is equally possible with our framework. For a detailed description of Test 1 and Test 2 refer to Appendix A.2.

Makelov et al. (2024) perform targeted edits on specific components of the IOI circuit for task 2, utilizing prior knowledge based on findings from Wang et al. (2023). In contrast, our framework avoids relying on prior knowledge. Instead, we leverage the attribute cross-sections to guide our editing choices. Each cross-section group is derived by computing attribution scores for a specific attribute $a_i$. Consequently, we focus on editing only the features associated with attribute $a_i$ when evaluating that particular cross-section group in Test 2. By making edits targeted at $a_i$, we expect the most significant downstream impact on the model's predictions for the corresponding cross-section group, providing a strong indicator of how effectively our supervised feature dictionaries can be applied to modify the features of $a_i$.

## 5  EXPERIMENTS

In this section we apply our SAGE framework to different tasks, models and feature distributions to demonstrate its scalability. We use two tasks: First, we apply our framework to the IOI task as a comparative baseline to the IOI evaluations of Wang et al. (2023). Second, we introduce an induction task across multiple feature distributions as a running example and apply our evaluation framework to it. We explain the setup of both tasks in detail in Appendix A.5. For the evaluations we use several open-source SAEs from the "sae_lens" library (Joseph Bloom, 2024). For Pythia 70M Biderman et al. (2023), we evaluate the "pythia-70m-deduped-res-sm" SAE with a dimension

of 32k, for GPT-2 Small Radford et al. (2019), we compare "gpt2-small-resid-post-v5-32k" with a dimension of 32k against "gpt2-small-resid-post-v5-128k" with a dimension of 128k. Lastly, for Gemma-2-2B et al. (2024) we compare the "gemma-scope-2b-pt-res-canonical" with dimensions 16k against the version with dimension 65k from Gemma-scope (Lieberum et al., 2024).

The induction task is based on the induction mechanism in LLMs (Olsson et al., 2022): induction heads recognize patterns in the input token sequence and predict tokens based on this pattern. The induction mechanism has several desirable properties: First, it enables applying the same task across all tested models for direct comparison as this task is implemented in most LLMs. Second, induction heads predict tokens based on patterns in the input sequence, allowing flexible token feature distributions for diverse evaluations.

## 5.1 Hyperparameters and Preprocessing

**Discover Cross-Sections**   For the cross-section discovery we have to decide which model components, component locations and how many patching examples to use for the calculation of the average attribution score. For the metric we choose logit difference between the *io* and *subject* token for the IOI task, and the *ind2* and *ind1* token for the induction task.

We choose to focus our evaluation on the attention heads and obtain attribution scores for edges between attention heads (attention head output as upstream node and the q, k and v input to attention heads as downstream nodes). We only consider locations in the model that are at the last attribute token position or later. This is because attributes like *order* can only be fully linearly represented after the last attribute token. Lastly, we choose to average the attribution scores across 250 task prompts that are sampled from the task datasets. For each task prompt we also sample one corrupted example differing in one attribute. Therefore, we have a total of 250 task prompt pairs to obtain the attribute cross-sections from.

**Filter Cross-Sections**   Next, we need to decide which cross-section to consider for the evaluation based on the obtained attribution scores. As the attribution scores are only approximations, we perform the following validation step to select the attribute cross-sections for the evaluation:

1. Select the top $n$ cross-sections for each cross-section group

2. Corrupt increasing subsets of the cross-section groups by patching the mean ablation of the upstream node in the input of the downstream node (edge mean ablation) and measure the change in the logit difference

3. For each cross-section group select the subset with the largest change in logit difference and drop cross-section groups that caused a change in logit difference below $\tau$

In our experiments we choose 300 cross-sections for GPT-2 Small and Pythia 70M, and top 1600 cross-sections for Gemma-2-2B. We dropped cross-section groups whose change in logit difference was below 60% of the mean change across all cross-section groups. Refer to Appendix A.3 to find the results of this selection procedure for each model and task.

**Supervised Dictionaries**   We train each supervised feature dictionaries on 10000 task prompts. We test how accurate our models completes the tasks across the sampled training data on 250 examples. For all experiments our models achieved at least 95% accuracy.

**Evaluations**   We perform the evaluations with Test 1 and Test 2 across 250 task examples, sampled from the test dataset of each task. For Test 2 we perform 0, 4, 8 and 16 edits. We also test the accuracy of our models in predicting the correct token for the test set and also achieve over 95% accuracy for all models and tasks.

## 5.2 SAGE on the IOI task

We first apply SAGE to the IOI task to demonstrates that SAGE reproduces the evaluation results of previous methods that relied on prior knowledge of the IOI circuit and trained SAEs for all circuit components, compared to SAGE, which only uses residual stream SAEs. The evaluation results for Test 1 and Test 2 on GPT-2 small are shown in Figure 2. Test 1 demonstrates that our

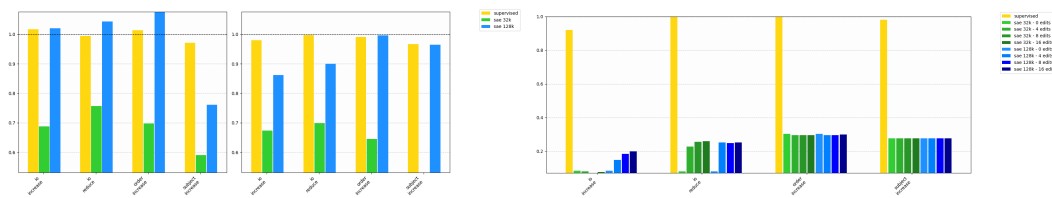

Figure 2: The figure to the left shows the scores for Test 1: (left) Sufficiency and (right) Necessity. The figure to the right is showing the results of Test 2.

supervised feature dictionary provides a strong approximation of the IOI features leading to excellent reconstructions of the cross-section activations, as it achieves sufficiency and necessity scores above 0.95. The 128k-dimension variant of the SAEs performs better than the 32k-dimension variant, but both fall short when compared to the supervised feature dictionary. In Test 2, the supervised feature dictionary demonstrated excellent sparse controllability with over 90% edit success across all cross-section groups. Similarly to the results of Makelov et al. (2024) for their full-distribution SAEs, we find that both SAEs struggle with sparse controllability, particularly when editing the *subject* and *order* features in the according cross-section groups, where even 16 feature edits fail to significantly influence the model's predictions. The 128k SAE demonstrates improvement for the *io-increase* group but still underperforms compared to the supervised dictionary, while the 32k variant does not improve.

Therefore, this experiment has shown that SAGE can discover high-quality, approximate ground truth feature dictionaries across the identified cross-sections for each feature of the IOI task, without relying on prior knowledge of the IOI task. We were able to evaluate SAEs on these cross-sections solely based on residual stream SAEs, while achieving comparable results to those reported by Makelov et al. (2024). Our results only differ in that editing the *subject* feature in the *subject-increase* group results in prediction flips. This discrepancy arises because Makelov et al. (2024) mainly focuses on editing the *subject* features in the inhibitor heads, whereas our method edits the *subject* feature in the cross-sections where patching increases the logit difference most. Thus, our edits of the *subject* feature in these cross-sections seem to effectively reduce the inhibitory signal instead of replacing it, causing the prediction to flip to a general token such as "the".

## 5.3 SAGE ON THE INDUCTION TASK

Next, we apply SAGE to the induction task, using the name feature distribution from the IOI task, across GPT-2 small, Pythia 70M, and Gemma-2-2B models. Our results validate that SAGE can scale to new tasks, with unknown circuits, new feature distributions, and state-of-the-art LLMs like Gemma-2-2B. Refer to Appendix A.4, for experiment results using the induction task with other feature distributions on GPT-2 Small. The evaluation results for Test 1 and Test 2 are shown in Figure 3. For all three models, the supervised feature dictionary achieves excellent approximations of the induction task features, with necessity and sufficiency scores exceeding 0.9. It also demonstrates strong sparse controllability, achieving over 80% success in feature edits, consistently outperforming both the 32k and 128k SAEs. The SAEs with larger dimensions generally outperform the smaller SAEs across most evaluation metrics. For GPT-2 small, we observe frequent prediction flips in the *ind1-increase*, *ind2-reduce*, and *order-increase* cross-section groups. The 128k-dimension SAE shows improvement with more feature edits in the *ind2-reduce* group, though it still vastly underperforms compared to the supervised dictionary. The 32k-dimension SAE also shows improvement with more edits in the *ind2-reduce* cross-section group, but not as good as the 128k variant. In the other groups more edits do not lead to improvements. For Pythia 70M, all cross-section groups lead to notable prediction flips. The 128k-dimension SAE improves with increasing edits for the *ind2-increase* and *ind2-reduce* groups, but struggles to match the performance of the supervised dictionary. The results for the *ind1* and *order* groups demonstrate minimal improvement with more feature edits, with both SAEs falling short of the supervised dictionary's accuracy across all groups. For Gemma-2-2B, we also see the primary prediction flips for the edits of the *ind2-reduce* cross-section group. However, increasing the number of edits has a more notable effect with 0% of correct

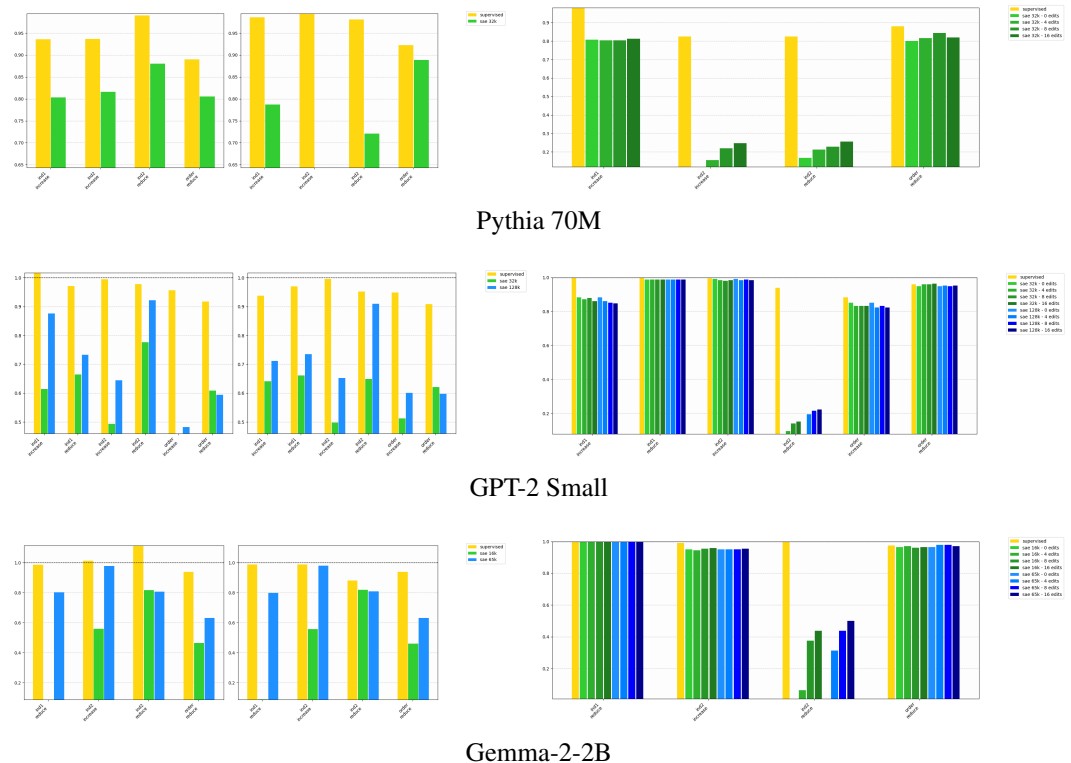

Figure 3: Evaluation results for the induction task on Pythia 70M, GPT-2 Small, and Gemma-2-2B. On the left are the scores for Test 1: (left) Sufficiency and (right) Necessity, and on the right are the results of Test 2.

predictions without edits, compared to 40% to 50% success rate for 16 edits. The 65k version out-performs the 16k version, and most notably the 65k version is able to perform the correct edit for 25% of the examples using only 2 edits compared to around 5% for the 16k variant. Thus, the 65k variant seems to find features in the activations that approximate the name features, used for the induction task, better than those of the 16k variant.

All in all, we have shown with this experiment that the SAGE framework can successfully scale SAE ground truth evaluations to new tasks, models and feature distributions, finding high quality supervised feature dictionaries, requiring only residual stream SAEs.

## 6 CONCLUSION AND FUTURE WORK

This paper introduces the SAGE framework, which enables to scale SAE ground truth evaluations to new models and task. To this end we have proposed a fully automated approach, while significantly reducing the training overhead of previous methods, using our novel projection-based reconstruction technique. We have demonstrated the scalability of SAGE by evaluating several SAEs on Pythia 70M, GPT-2 Small, and Gemma-2-2B with a new task using different feature distributions. Future work includes expanding task diversity by incorporating tasks with more high-level features and covering larger feature distributions. The current approach is constrained by the effectiveness of at-tribution patching in identifying attribute cross-sections. Thus, future research on improved discov-ery methods would further enhance our technique. Another potential extension is the incorporation of additional activation disentanglement methods beyond SAEs, such as PCA or Distributed Align-ment Search, to provide a more comprehensive evaluation. In summary, SAGE enables large-scale SAE evaluations on realistic tasks, marking a significant advancement toward generalizable SAE assessments in interpretability research.

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

# A  APPENDIX

## A.1  SUPERVISED FEATURE DICTIONARIES

Makelov et al. (2024) give two primary ways to derive supervised feature dictionaries at cross-sections, MSE feature dictionaries and mean feature dictionaries. We use MSE feature dictionaries, as they are recommended in the general case. They can be obtained using the following method:

**Step 1: Define the Dataset and Model Components**  Let $\mathcal{D} = \{p_k\}_{k \in N}$ be the dataset of input prompts. Let $\mathbf{a}(p_k) \in \mathbb{R}^d$ be the activation of a model component for prompt $p_k$. Let $\{a_i\}_{i \in I}$ be the set of attributes that describe the task, where each attribute $a_i : \mathcal{D} \to S_i$ is instantiated with a specific value from $S_i$ for each prompt in the dataset.

**Step 2: Center the Activations**  Compute the mean activation over the dataset:

$$\bar{\mathbf{a}} = \frac{1}{N} \sum_{k=1}^{N} \mathbf{a}(p_k).$$

Center the activations by subtracting the mean:

$$\tilde{\mathbf{a}}(p_k) = \mathbf{a}(p_k) - \bar{\mathbf{a}}.$$

**Step 3: Encode the Attribute Values**  For each attribute $a_i$ and value $v \in S_i$, define an indicator function $\mathbb{1}_{a_i(p_k)=v}$ that equals 1 if the attribute $a_i$ of prompt $p_k$ equals $v$, and 0 otherwise.

Construct the attribute matrix $C \in \mathbb{R}^{N \times m}$ where $m = \sum_{i=1}^{I} |S_i|$. Each column of $C$ corresponds to an indicator function $\mathbb{1}_{a_i(p_k)=v}$ for a particular $a_i$ and $v$.

**Step 4: Solve the Least-Squares Problem**  The goal is to learn feature vectors $\mathbf{u}_{a_i=v} \in \mathbb{R}^d$ for each attribute $a_i$ and value $v$ by solving the following least-squares problem:

$$\underset{\mathbf{u}_{a_i=v}}{\arg\min} \frac{1}{N} \sum_{k=1}^{N} \left\| \tilde{\mathbf{a}}(p_k) - \sum_{i \in I} \mathbf{u}_{a_i=a_i(p_k)} \right\|_2^2$$

In matrix form, the problem is:

$$\min_{U} \frac{1}{N} \left\| \tilde{A} - CU \right\|_2^2$$

where:

- $\tilde{A} \in \mathbb{R}^{N \times d}$ is the matrix of centered activations.
- $C \in \mathbb{R}^{N \times m}$ is the attribute matrix.
- $U \in \mathbb{R}^{m \times d}$ is the matrix of feature vectors, with rows $\mathbf{u}_{a_i=v}$.

**Step 5: Solve the System**  The solution to this least-squares problem is given by:

$$U^* = \left(C^T C\right)^+ C^T \tilde{A}$$

where $\left(C^T C\right)^+$ is the Moore-Penrose pseudoinverse of $C^T C$.

**Step 6: Obtain the Final Feature Vectors**  Each row of $U^*$ corresponds to a feature vector $\mathbf{u}_{a_i=v}$ for a specific attribute $a_i$ and value $v$.

## A.2  EVALUATION TESTS

In this section, we describe two of the evaluation tests proposed by Makelov et al. (2024) that we use to show that the ground truth features obtained with our approach, can be used for SAE evaluations.

**Test 1: Sufficiency and Necessity**    This test evaluates whether the feature dictionary's reconstructions are sufficient and necessary for the model to perform the task. For sufficiency, the test involves replacing the activations of a cross-section group with their reconstructions from the SAE and the supervised feature dictionary. Then the effect on the logit difference is measured, by calculating the average logit difference across all clean runs $L_c$, all runs where the activations of the relevant components are exchanged with their reconstruction $L_s$, and all runs where we patch in the mean activations, as an in-distribution baseline of a reconstruction that does not capture any task-relevant features $L_m$. A score of the sufficiency of the reconstructions can then be obtained as:

$$\frac{|L_s - L_m|}{|L_c - L_m|}$$

For necessity, the test involves assessing whether the features captured by the feature dictionary are necessary for the model performance. To do this, instead of patching in reconstruction, we aim to delete the relevant features from the components in our set by replacing each cross-section activation $\mathbf{a}$ with $\bar{\mathbf{a}} + (\mathbf{a} - \hat{\mathbf{a}})$. We obtain the average logit difference of the output logits of all clean runs $L_c$, all runs where the relevant activations are replaced with the difference between the clean activations and their reconstructions $L_n$, and the in-distribution baseline $L_m$. A score of the necessity of the reconstructions can be obtained as:

$$1 - \frac{|L_n - L_m|}{|L_c - L_m|}$$

This formulation directly captures how much of the original information is lost when the reconstructed activations are removed, providing a measure of how necessary the dictionary features are for the model's task performance.

**Test 2: Sparse Controllability**    This test evaluates the extent to which features in the learned dictionary can be used to sparsely control the model's behaviour by editing specific attributes of the input prompts. Let task prompts $p_s$ and $p_t$ differ in exactly one attribute. Let $\mathbf{a}(p_s)$ and $\mathbf{a}(p_t)$ represent the activations of cross-sections of the task for each task prompt. Let $\hat{\mathbf{a}}(p_s)$ and $\hat{\mathbf{a}}(p_t)$ represent the respective reconstructions of the activations using the feature dictionary (SAE or supervised). We aim to determine whether modifying each activation $\mathbf{a}(p_s)$ using the features used for the reconstructions $\hat{\mathbf{a}}(p_s)$ and $\hat{\mathbf{a}}(p_t)$ across all relevant cross-sections can flip the prediction of the model to what it would predict under the activations $\mathbf{a}(p_s)$. Formally, the problem is expressed as:

$$\min_{R \subseteq S, A \subseteq T, |R \cup A| \leq k} \left\| \mathbf{a}(p_s) - \sum_{i \in R} \alpha_i \mathbf{u}_i + \sum_{i \in A} \beta_i \mathbf{u}_i - \mathbf{a}(p_t) \right\|_2^2,$$

where $S$ and $T$ are the sets of active features $\mathbf{u}_i$ in the reconstructions $\hat{\mathbf{a}}(p_s)$ and $\hat{\mathbf{a}}(p_t)$ respectively, and $\alpha_i, \beta_i > 0$ are their coefficients. The problem of finding the optimal sparse edit for the SAE features is NP-complete, as it can be reduced to the *Subset Sum* problem (Makelov et al., 2024).

The ground truth edit is replacing $\mathbf{a}(p_s)$ with $\mathbf{a}(p_t)$. Editing with the supervised feature dictionary works by subtracting the feature $\mathbf{u}_{a_j = s_v}$ from $\mathbf{a}(p_s)$ and adding $\mathbf{u}_{a_j = s_t}$, where $a_j$ is the attribute we wish to edit and $s_v$ is the value it takes in $p_s$ and $s_t$ is the value for the attribute in $p_t$.

As already explained, finding the optimal edit for the SAE setting is NP-complete. Makelov et al. (2024) use a greedy algorithm to approximate the optimization problem by exchanging a fixed number of features, such that the activation $\mathbf{a}(p_s)$ gets closer to $\mathbf{a}(p_t)$:

1. Obtain the features $\mathbf{u}_1^s, u_2^s, ..., \mathbf{u}_m^s$ and their coefficients $c_1^s, c_2^s, ..., c_m^s$ that we obtain from the reconstruction of $\mathbf{a}(p_s)$. Similarly obtain and $\mathbf{u}_1^t, u_2^t, ..., \mathbf{u}_m^t$ and the coefficients $c_1^t, c_2^t, ..., c_m^t$ for $\mathbf{a}(p_t)$.

2. In the next step, we iterate over all pairs of features and apply the edit $\mathbf{a}'(p_s) = \mathbf{a}(p_s) - c_i^s \cdot \mathbf{u}_i^s + c_j^t \cdot \mathbf{u}_j^t$ and calculate the distance between $\mathbf{a}'(p_s)$ and $\mathbf{a}(p_t)$.

3. We select the edit that makes both activations most similar and repeat that procedure for the specified number of edits.

## A.3 CROSS-SECTION GROUPS: SELECTION RESULTS

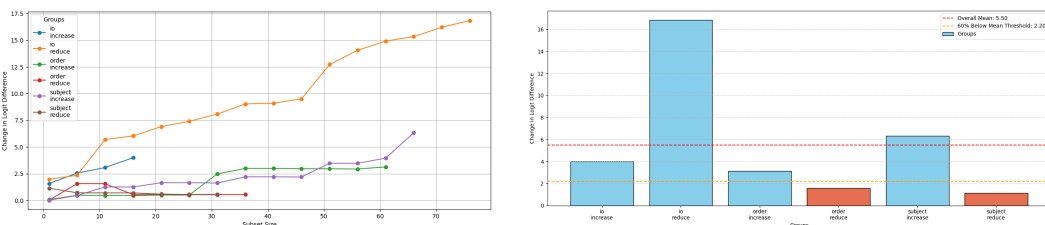

GPT-2 Small & IOI task

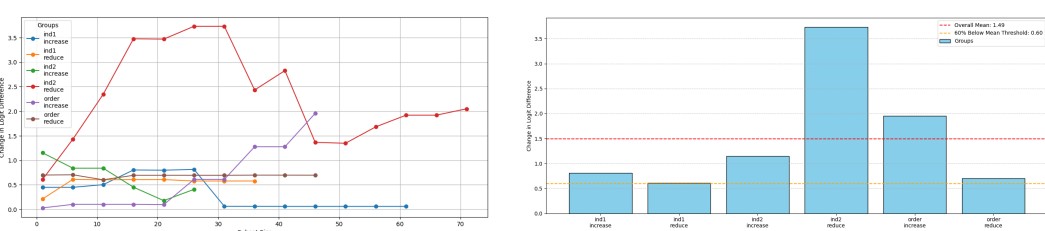

GPT-2 Small & induction task

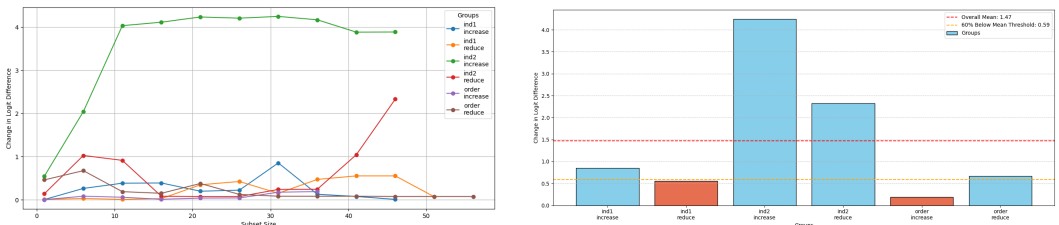

Pythia 70M & induction task

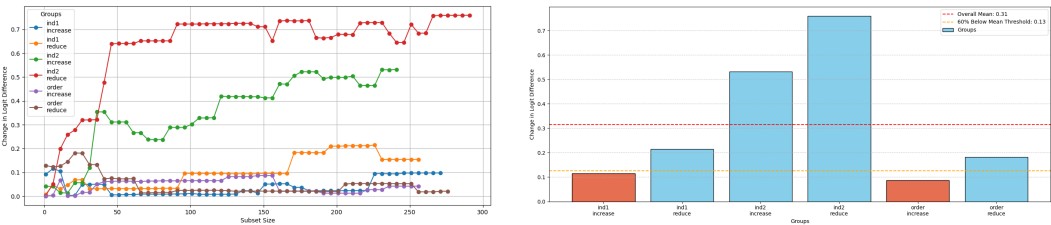

Gemma-2-2B & induction task

Figure 4: Shown are the results of the selection procedure described in the Experiment section. The plot on the left shows on the x-axis the subset size of the different cross-section groups. On the y-axis it shows the change in logit difference, when mean-ablating the cross-sections of each subset. The plot on the right shows which cross-section groups were filtered out (shown in red) because their change in logit difference was below 60% of the average change in logit difference across all cross-section groups.

## A.4 DIFFERENT FEATURE DISTRIBUTIONS

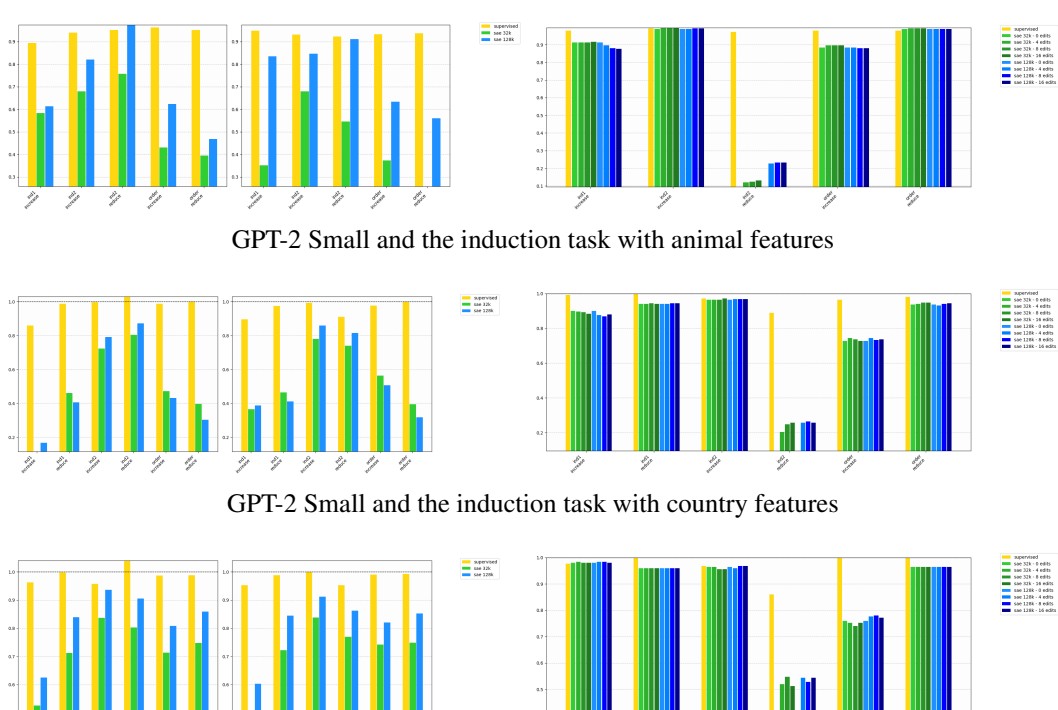

GPT-2 Small and the induction task with animal features

GPT-2 Small and the induction task with country features

GPT-2 Small and the induction task with number features

Figure 5: Results of Test 1 and Test 2 for the induction task, with different feature distributions (10 tokens of the according feature category each).

## A.5 EVALUATION TASKS

**IOI Task** We set up the IOI task to be defined over the attributes *indirect object (io)*, *subject*, and *order*. As values for the *io* and *subject* attribute we choose a set of ten single-token names from the IOI name distribution. As values for the *order* attribute we choose $\{abb, bab\}$. To initialize IOI prompts, we use three IOI templates (slightly altered from Wang et al. (2023) to align the token positions). The following two templates with the according *io* and *subject* orderings are used to train the supervised feature dictionaries:

```
Then, {io} and {subject} had a long argument.  {subject} gave a drink to

Then, {subject} and {io} had a long argument.  {subject} gave a drink to

Then, {io} and {subject} went to the store.  {subject} gave an apple to

Then, {subject} and {io} went to the store.  {subject} gave an apple to
```

To run the evaluation tests, we use the following template:

```
Then, {io} and {subject} went to the cafe.  {subject} gave the cake to

Then, {subject} and {io} went to the cafe.  {subject} gave the cake to
```

**Induction Task** For the induction task we use the following algorithm to sample token patterns for which the model can perform induction with low cross-entropy:

---

**Algorithm 1** Induction Sequence Sampling

---

**Require:** Model $M$, Vocabulary $V$, threshold $\tau > 0$
**Ensure:** Induction sequence $s$
1:  CE $\leftarrow \infty$              $\triangleright$ Initialize cross-entropy to a large value
2:  **while** CE $> \tau$ **do**
3:    $r \leftarrow$ `sequence of` $n$ `tokens from` $V$
4:    $T \leftarrow m$ `target tokens from` $V$
5:    CE $\leftarrow []$           $\triangleright$ Initialize cross-entropy list
6:    **for all** $t \in T$ **do**
7:     $x \leftarrow r + t + r$        $\triangleright$ Create induction sequence
8:     $\ell \leftarrow M(x)$        $\triangleright$ Compute logits for sequence
9:     CE $+= -\log(\texttt{softmax}(\ell)[t])$     $\triangleright$ Add cross-entropy
10:    **end for**
11:    **if** mean(CE) $\leq \tau$ **then**
12:     **return** $r$
13:    **end if**
14: **end while**

---

We set up the induction task with three attributes: *ind1*, *ind2*, and *order*. The attribute *ind2* is the output of the induction task, *ind1* is the previous token for *ind2* and used to define two separate orderings that are captured with the attribute *order*. Therefore, the task consists of two token features *ind1* and *ind2*, for which a feature distribution can be freely defined, and a high level *order* attribute. Note that this is different from the IOI *order* attribute, as the ordering for the induction task differs significantly. Based on these attributes we define the following prompt templates for the induction task:

`{seq} {ind2},{ind1},{ind2},{ind1} {seq} {ind2},{ind1}, → {ind2}`
`{seq} {ind1},{ind1},{ind2},{ind2} {seq} {ind1},{ind1}, → {ind2}`

Then we can sample two training sequences to instantiate $\{seq\}$, and one test sequence.

