# OpenReview forum: "SAGE: Scalable Ground Truth Evaluations for Large Sparse Autoencoders"
_ICLR.cc/2025/Conference — ICLR 2025 Conference Withdrawn Submission_

### Official Review · Reviewer_Fr97 · 2024-11-02

**Soundness:** 2
**Presentation:** 1
**Contribution:** 2
**Rating:** 5
**Confidence:** 2

**Summary:**

The paper proposes a scheme to automate SAE evaluations for large models/complex tasks and for situations where ground truth features are not known.

There are several claimed contributions:

A novel (as far as I can tell) approach to identify cross sections by leveraging clean and corrupted examples w.r.t the attribute of interest.

“A novel projection-based reconstruction using residual stream SAE”

A modification to the supervised feature dictionary approach of Makelov (2024) in which they instead “propose to compute weights to minimize the MSE loss between the reconstruction and the activation:”

Taken together, these innovations allow SAE evaluations to be “scaled,” apparently meaning w.r.t tasks with a lack of ground truth features or increased model size.

**Strengths:**

The chosen task is important and highly relevant.

**Weaknesses:**

Philosophically, I don’t understand the point of this paper. If I am reading it correctly, and it’s possible I am not, the idea is to generate pseudo-ground truth features via “supervised feature dictionary learning” for cross-sections corresponding to features of interest (which are also identified automatically). Then it is claimed these features can be used to evaluate SAEs. But this seems to me to defy the whole point of SAE evaluation in the first place, which is to isolate network components which are interpretable to humans.

The presentation of the method is also generally pretty confusing because it’s entangled with the presentation of background and related work. I also feel like it’s too abstract and needs a lot more concrete examples and figures to be grounded effectively.

The experimental results are hard to read, both due to the small font and the chart-style presentation.

**Questions:**

It would be great if you could address weakness 1. I also think the presentation of the method and results should be made much less abstract, i.e. with some concrete examples.

---

### Official Review · Reviewer_scNh · 2024-11-03

**Soundness:** 2
**Presentation:** 1
**Contribution:** 1
**Rating:** 3
**Confidence:** 3

**Summary:**

The paper introduces an evaluation framework to evaluate sparse autoencoders called SAGE. The approach consists of attribute cross-sections, supervised feature-dictionary reconstruction and projection-based reconstruction.  The framework is evaluated on novel tasks on Pythia 70M, GPT-2 small and Gemma-2-2B.

**Strengths:**

The paper is dealing with a problem that Is of interest to the ICLR community.

**Weaknesses:**

The main weakness of this paper Is it’s presentation, it is very difficult to get through the manuscript and understand the introduced evaluation framework. One would expect to have a clear answer to the following three questions after reading the abstract and the intro: (1) what is done, (2) why it is done, and (3) how it is done. However, after reading the paper multiple times, the reviewer still struggles to find the responses to the above-mentioned questions.  For details, please see Questions box.

The current paper presentation makes it difficult to assess the papers’ novelty, impact and significance of the results.

**Questions:**

Abstract has some contradictions in it, it seems that the paper is about an evaluation framework, however, at the same time it seems like there are some model improvements, e.g., ‘introduce a novel reconstruction method’.

Abstract. The validation is also unclear, the authors say that they validate the results on  novel tasks on 3 datasets, however, it is unclear what is the conclusion of these validations. Could the authors clarify this?

Could the authors clarify the novelty of the introduced approach?

Intro 2nd paragraph – the connection between the interpretability of LLMs and SAEs is not clear – could the authors clarify the connection?

Intro 3rd paragraph – it is not clear what circuit discovery has to do with evaluation of SAE, could the authors clarify this?

Intro: Could you please clarify what is meant by large frontier model?

Intro: What the authors mean by linearly represented features?

Intro: The use of manual experimentation is unclear.

Intro: What the authors mean by sublayer activations and residual stream SAE?

Please make all the legends in all figures bigger.

---

### Official Review · Reviewer_8kKN · 2024-11-03

**Soundness:** 3
**Presentation:** 2
**Contribution:** 2
**Rating:** 3
**Confidence:** 3

**Summary:**

Authors consider the question of evaluating sparse auto-encoder
models, which are studied for interpreting large scale models. They
start from a recent article and build on top of it. The previous
article already proposed a strategy for evaluating SAE models using
supervised dictionary features, extracted for a given task and points
(cross-sections) in a language model. Here, this work is extended by
first identifying the cross-section automatically and then using a
different training strategy for the SAEs. This way, authors claim that
the evaluation can be done for large-scale LLMs.

**Strengths:**

+ Interpretation of components in LLMs is a highly relevant topic.
+ SAEs is an interesting direction that recently attracted attention.
+ Automated identification of cross-sections is a step forward. The
  previous work relied on prior knowledge and thus was limited on what
  it could be applied. Using automated discovery of cross-sections,
  this limitation is alleviated.

**Weaknesses:**

- overall, I think the problem definition and the description can be
  better written. Author rely heavily on the previous work, i.e.,
  Makelov et al., for their description. However, the current article
  on its own is not trivial to understand.
- It is unclear how the supervised dictionaries are extracted in this
  work and whether this is something new. Makelov already discusses in
  appendix A.6 the MSE estimation of the supervised
  dictionaries. Furthermore, in that model $u$ vectors are extracted
  using MSE. Here, it is unclear how $u$ vectors are
  extracted. Equation 5 and 6 only describes the weighting of the
  vectors.
- Projection-based reconstruction with residual streams is not really
  justified in my opinion. More specifically, this section makes me
  question the goal of this work. In Makelov et al.'s work, authors
  aim to evaluate SAEs, thus the cost of training SAEs is not really
  an issue. If the goal here is the same, why is the training of SAEs
  all of a sudden an issue? I am not sure if the motivation for
  reducing the training cost and the development for the residual
  stream is clear at all.
- Certain parts of the article are put in the Appendix, e.g.,
  A.5. This makes me wonder whether given the page limitations of
  conferences, a journal would be a better venue for this work.
- Discovery of the cross-sections is the main novelty in my opinion,
  however, this is also a part that is not trivial to evaluate and
  assess the accuracy. Authors should perhaps discuss this a bit
  further. Is there a way to really show that this is indeed a good
  method?

Minor:
- the last $x_{corr}$ on line 264 is most likely $x_{clean}$.

**Questions:**

- I suggest improving the writing, especially in the method section.
- The motivation for accelerating training of SAEs need to be better
  justified.
- How can one assess the accuracy of the cross-section discovery?

---

### Official Review · Reviewer_17D3 · 2024-11-03

**Soundness:** 3
**Presentation:** 2
**Contribution:** 3
**Rating:** 5
**Confidence:** 3

**Summary:**

This work tries to scale up the feature evaluation problem for large sparse autoencoders (SAEs) in large language models, towards interpretable AI. The main contribution of the work is a new reconstruction approach for task-specific features with large training reduction of the SAEs. The overall method and experimental results look fine.

**Strengths:**

Studying intrinsic features that are interpretable in large language models is an interesting and important direction.
The  large sparse autoencoders (SAEs) and indirect object identification (IOI) look promising. This is a relatively new direction and the proposed method and findings make a good attempt in this direction. The proposed reconstruction method seems to be sound.

**Weaknesses:**

This is a paper difficult to read. It is mainly based on two prior works,

[1] . "Interpretability in the wild: a circuit for indirect object identification in GPT-2 small", Wang et al. 2023,
which defines the problem of indirect object identification (IOI) with a discovery of circuit; and

[2]  "Towards Principled Evaluations of Sparse Autoencoders for Interpretability and Control", Makelov et al.
that gives a framework for supervised learning of sparse autoencoders (SAEs) for the indirect object identification (IOI) task.

I am not very familiar with the IOI and SAE literature, which seems to be a recently proposed problem in large language models. [1] and [2] were written in a relatively clean and clear way for readers to quickly understand the motivation and problem definition.

However, the paper in study is much harder to read than [1] and [2]. This may be also due to my unfamiliarity to this direction. This paper seems to make an assumption of having the readers with all the prior knowledge about IOI and SAEs by quickly throwing out the jargon words without much explanation.

**Questions:**

My main concern regarding this paper is its writing and presentation.

---

### Note · Authors · 2024-12-04

I have read and agree with the venue's withdrawal policy on behalf of myself and my co-authors.